# Varicocele, Functional Foods and Nutraceuticals: From Mechanisms of Action in Animal Models to Therapeutic Application

**DOI:** 10.3390/ijms232416118

**Published:** 2022-12-17

**Authors:** Herbert Ryan Marini, Antonio Micali, Domenico Puzzolo, Letteria Minutoli, Pietro Antonuccio

**Affiliations:** 1Department of Clinical and Experimental Medicine, University of Messina, 98125 Messina, Italy; 2Department of Human Pathology of Adult and Childhood, University of Messina, 98125 Messina, Italy; 3Department of Biomedical and Dental Sciences and Morphofunctional Imaging, University of Messina, 98125 Messina, Italy

**Keywords:** varicocele, fertility, inflammation, animal studies, Mediterranean diet, functional foods, nutraceuticals

## Abstract

Varicocele is one of the main causes of infertility in men, thus representing an important clinical problem worldwide. Inflammation contributes mainly to its pathogenesis, even if the exact pathophysiological mechanisms that correlate varicocele and infertility are still unknown. In addition, oxidative stress, apoptosis, hypoxia, and scrotal hyperthermia seem to play important roles. So far, the treatment of varicocele and the care of the fertility-associated problems still represent an area of interest for researchers, although many advances have occurred over the past few years. Recent experimental animal studies, as well as the current epidemiological evidence in humans, demonstrated that many functional foods of natural origin and nutraceuticals that are particularly abundant in the Mediterranean diet showed anti-inflammatory effects in varicocele. The aim of the present narrative review is to mainly evaluate recent experimental animal studies regarding the molecular mechanisms of varicocele and the state of the art about possible therapeutic approaches. As the current literature demonstrates convincing associations between diet, food components and fertility, the rational intake of nutraceuticals, which are particularly abundant in foods typical of plant-based eating patterns, may be a reliable therapeutic supportive care against varicocele and, consequently, could be very useful in the cure of fertility-associated problems in patients.

## 1. Introduction

Varicocele is a pathological venous dilation of an internal testicular vein and of the pampiniform plexus, which is located in the spermatic cord, inducing blood reflux [1]. It is considered an important cause of male infertility, being present in nearly 15–20% of the male population; in males with primary infertility, its frequency is about 35%, while in males with secondary infertility, it is 70–80% [2]. However, the relationship between varicocele and the failure in adult fertility remains unclear [3]. Normally, varicocele is observed on the left side (90% of cases), while only in 10%, it occurs bilaterally. In less than 1% of patients, varicocele is observed on the right side [4]. The presence of left-sided varicocele was considered based on the peculiar anatomic arrangement of the left internal spermatic vein, which drains, forming a nearly right angle, into the left renal vein. On the contrary, the right internal spermatic vein typically ends into the inferior vena cava, forming an acute angle [5].

Moreover, the left internal spermatic vein is about 8–10 cm longer than the right one. This, in addition to the anatomical arrangement of the blood vessel, causes an increase of the hydrostatic pressure on the left side, and consequently, the dilation of the pampiniform plexus (hydrostatic pressure theory) [5]. Among the causes of varicocele formation, an important role is attributed to the incompetence or even absence of valves of the spermatic veins, which causes an uninterrupted retrograde flow from the internal spermatic vein into the pampiniform plexus (valvular mechanisms theory) [6]. A third proposed mechanism in varicocele formation is the so-called “nutcracker effect”, occurring when the left renal vein is compressed in the angle formed by the aorta and, most commonly, the superior mesenteric artery [7]. Consequently, pressure elevation induced by venous stasis is transmitted to the left internal spermatic vein and to the pampiniform plexus (nutcracker phenomenon theory) [8].

Even if varicocele is usually asymptomatic [9], the patient may complain of scrotal swelling and weight sensation, even if a low percentage suffer from scrotal pain [10]. It has been demonstrated that the disease is able to induce reduced sperm motility, low sperm count, anomalous sperm structure, lower testicular volume and impairment of Leydig cell function [2].

Consequently, the damage of testicular function is linked to physio-pathological events, including testicular hypoxia and hyperthermia, inflammation, oxidative stress, hormonal imbalance and cellular death [2,11,12,13]. The molecular links between varicocele and infertility are still unclear, as multiple mechanisms are involved.

A correct diagnosis of varicocele is particularly important to avoid a worsening of the pathology and to be able to combine traditional surgery with treatment with natural substances capable of modulating the pathogenetic mechanisms involved in the disease, with particular regard to the inflammatory cascade (Figure 1).

## 2. Materials and Methods

We consulted the following databases: MEDLINE-PubMed and Google Scholar. The keywords “varicocele and pathophysiology”, “varicocele and epidemiology”, “varicocele and animal models”, “varicocele and inflammation”, “varicocele and inflammatory nutraceuticals”, and “varicocele and diet” were selected for data extraction, also including the main references of the chosen articles. In the first search, we found 370 articles, which were selected by reading their titles and abstracts, when present. If the articles were considered relevant for the review, the full text was read. We included reviews and original papers from 1981 to 2022, but only 71 were chosen on the basis of their topic. We excluded some papers dealing with clinical features on varicocele, as our review was based mainly on animal models. Other papers were excluded, as they examined only different features of the pathophysiology of varicocele, such as oxidative stress and apoptosis, or because the considered molecules showed effects different from anti-inflammatory, which was the main topic of our review.

## 3. Varicocele Experimental Models

Even if varicocele is almost exclusively observed in humans, owing to the upright posture, animal models have been suggested, all based on the incomplete occlusion of the left renal vein after the insert of the internal spermatic vein. In particular, in rats, the first surgical approach was proposed by Saypol et al. [14]. Since then, many authors performed this technique, known as the classical approach, even if with slight modifications.

After induction of anesthesia, through an abdominal midline incision, the left renal vein, the inferior vena cava, and the left spermatic vein are identified, and a clamp is passed behind the left renal vein just distal to the spermatic vein insertion. A silk ligature is loosely placed around the left renal vein at this site and a rigid probe is placed on the left renal vein; then, the ligature is tied around the vein over the top of the probe. The latter is withdrawn, and the vein is allowed to expand to the limits of the ligature (approximately half of its original diameter). Therefore, renal and spermatic veins dilate immediately [13].

Besides the classical method of varicocele induction, another microsurgical model was proposed in rats [15]. In addition to the partial ligation of the left renal vein, also the spermatic vein branches to the common iliac vein are fully ligated under a surgical microscope, thus apparently improving the effectiveness of the rat varicocele model. However, considering the results obtained and compared between this and the classic method, this latter is still the common choice in order to induce varicocele in animal models, especially in rat models [1]. Specifically, it has been suggested that the latest surgical approach, in order of time, allows to better characterize the testicular immune response, including the production of anti-sperm antibodies and inflammatory factors as well as the robust activation of inflammatory pathways in varicocele [1].

## 4. Varicocele and Inflammation

Varicocele triggers inflammation through infiltration and activation of macrophages and lymphocytes, which leads to higher quantities of inflammatory cytokines [16]. Among the cytokines, some have an anti-inflammatory action, such as Interleukin (IL)-37, which is secreted from macrophages in semen fluid and is able to suppress excessive inflammatory responses, particularly the pro-inflammatory activity of IL-18. Both of these cytokines were observed to be increased in the seminal plasma of varicocele patients; therefore, the main function of IL-37 is to enable a negative feedback mechanism to improve sperm motility [17].

Among the other cytokines involved in the testicular tissue during varicocele, IL-6 and tumor necrosis factor-α (TNF-α) play an important pro-inflammatory role. As for the IL-6, Leydig and Sertoli cells and testicular macrophages are considered at the site of secretion [18]. Its overexpression and/or over production causes inflammatory reactions and triggers morphological changes in the testis structure [19], consisting in a reduction of spermatogonia differentiation [20]. In addition, increased concentrations of IL-6 induce an inhibition of testosterone synthesis [18].

TNF-α is another cytokine implicated at physiological levels in diverse functional activities in testes, such as androgen receptor expression in Sertoli cells and lactate providing to spermatid and spermatozoa through Sertoli cells [18]. However, high quantities of TNF-α trigger the inflammatory cascade, acting on the infiltration and/or activation of peritubular immune cells [21] and causing reduced spermatozoa quality, involving count, motility, and structure [22]. In varicocele, both high levels and marked positivity of TNF-α were demonstrated with ELISA assessment and immunohistochemical staining [18].

In addition, the increased production of TNF-α, together with augmented Transforming Growth Factor (TGF)-β, a pleiotropic cytokine with strong regulatory and inflammatory activity, impairs the normal organization of the blood-testis barrier (BTB) [23]. In fact, as a result of the impaired BTB a down-regulation of claudin-11, a membrane protein component of tight junctions, was observed in a rat model of varicocele [23]. It is well known that the integrity of the BTB and its regular reorganization is necessary for a normal spermatogenesis, as it divides the seminiferous tubule into a basal compartment, where spermatogonia and primary spermatocytes are present, and an adluminal compartment, where all the other germinal cells are located [24]. BTB is made by adjacent Sertoli cells, connected by adherens and tight junctions; its integrity is based on the regular expression of transmembrane proteins, such as occludin, claudin-11 and N-cadherin [25]. In varicoceles, the activation of inflammatory pathways can induce the destruction of the BTB, thus producing male infertility [26].

Another mediator involved in varicocele-induced inflammation can be considered cyclooxygenases (COX), the enzymes which convert polyunsaturated fatty acids and arachidonic acid to prostaglandins (PGs). In particular, COX2 induces the production of PGE_2_, thus further amplifying inflammatory reactions. In testis, COX2 is constitutively expressed in germinal and in Sertoli and Leydig cells [27]. Furthermore, the increased presence of polymorphonuclear and mononuclear immune cells can induce an overexpression of iNos, IL-1, and IL6- in varicoceles. In varicocele-impaired spermatogenesis of infertile men, an increased expression of COX2 was demonstrated; thus, it was proposed as a biomarker for varicocele-induced infertility [27].

During inflammation, an activation of inflammasomes was also observed. These multiprotein complexes are formed by different members of nucleotide oligomerization domain (NOD)-like receptor family pyrin domain (NLRP1, NLRP2, NLRP3) and absent in melanoma (AIM), but in varicocele pathogenesis, the NLRP3 inflammasome was shown to play an important role. Once activated, it splits procaspase-1 to caspase-1, which converts the pro-interleukin-1β and 18 to their active forms. The consequent increase of these pro-inflammatory cytokines in the testis could cause damages in testicular function, spermatogenesis and androgen production [13,28].

In fact, varicoceles are also characterized by a decrease of serum testosterone levels, which is related to a down-regulation of the Steroidogenic acute regulatory protein (StAR) protein, whose role in promoting the transport of cholesterol into the mitochondria for testosterone synthesis was demonstrated [29]. The increase of inflammatory cytokines is able to downregulate the expression of StAR in testicular tissue, thus reducing testosterone production [30] (Figure 1).

## 5. Dietary Habits and Their Effects in Varicocele

The care of fertility-associated problems in varicoceles is strongly concentrated over the last few years, when the role of inflammation was demonstrated, owing to the increased levels of cytokines correlated with lower sperm motility, augmented sperm necrosis, and altered spermatogenesis [31].

Consequently, the dietary pattern and the components of the diet and nutrients have been studied, particularly from a molecular point of view, as possible bases of sperm function and/or fertility [32]. The “Western diet” (WD), which is characterized by a high intake of industrially handled foods, rich in animal proteins, simple carbohydrates, trans and saturated fats, and poor in dietary fiber and essential unsaturated fatty acids has been related to an increased risk of infertility [33,34]. On the other side, the Mediterranean diet (MD), rich in monounsaturated fatty acids, fibers, and antioxidants and low in saturated fat, appears to be protective against infertility [35,36,37]. Curiously, another dietary plant-based pattern, “the vegetarian diet” (VD), similar in its composition to MD, but eliminating meat and meat products, poultry, seafood, and flesh from any other animal, shows controversial data [38]. From a molecular point of view, the WD pattern contributes to infertility in varicoceles through a series of pro-inflammatory actions. First, it facilitates obesity, which causes insulin resistance, leading, in turn, to the development of oxidative stress and exaggerated inflammatory response, thus altering reproductive pathways and sperm function [39,40,41]. Second, it was demonstrated that WD induces a condition of hyperleptinemia that, by facilitating the chronic pro-inflammatory state in the testicular microenvironment, increases the level of ROS, which, in turn, is responsible for the decrease in sperm quality [40,42,43]. Finally, the surplus of fat tissue results in an increased activity of aromatase and in a consequent decrease of testosterone levels, resulting in lower sperm production; a finding that is also supported by a marked oxidative stress, as well as by a deep mitochondrial dysfunction [44,45].

By contrast, MD provides a low level of saturated and trans fatty acids with adequate levels of nutrients, such as omega-3 fatty acids, antioxidant molecules, and vitamins. Moreover, higher intakes of fruit, cereals, and vegetables, as well as the consumption of extra virgin olive oil, were positively related to sperm motility and concentration [46,47,48]. Thus far, regarding the latter, extra virgin olive oil, which is the main source of monounsaturated fatty acids, may modify the sperm membrane lipid composition through the reduction of ROS and the restoration of mitochondrial function [49]. Moreover, MD is characterized by a reduction in omega-6 and an increase of omega-3 fatty acids, which have been associated with an improvement of sperm energetic metabolism [49].

Surprisingly, the role of a VD in the preservation of sperm quality is controversial. As a matter of fact, it was indicated that this dietary pattern reduced sperm concentration and spermatozoa motility, probably because the occurrence of environmental estrogenic compounds in the VD, which could have a negative effect on sperm parameters [50].

Overall, the differential impacts of WD, MD, and/or VD on male fertility depend on the amount and quality of the nutrients introduced [32]. Therefore, a strong adherence to a healthy dietary pattern based mainly on plant foods and fish is positively correlated with indicators of sperm quality.

Thus far, as suggested in the accurate review by Ferramosca and coworkers [32], diets rich in saturated fatty acids and low in polyunsaturated fatty acids or those with an unbalanced omega-6/omega-3 polyunsaturated fatty acids ratio negatively affect sperm quality, whereas diets characterized by an unsaturated fatty acid supplementation ameliorate sperm quality. Furthermore, an excess of simple carbohydrates negatively affects sperm function, and a low-protein diet and the deficiency of some specific amino acids could represent a potential risk factor for the development of male infertility. Then, these macronutrients, all acting on oxidative stress and mitochondrial dysfunction causing exaggerated inflammatory state, as well as through the strategical modulation of testosterone levels acting on the neuro-immune-endocrine system, could play a key role in the improvement of sperm quality. In this context, as well summarized in the systematic review by Salas-Huetos and coworkers [51], since all observational studies up to now conducted in humans may prove associations but not causation, the associations between dietary patterns, foods and nutrients need to be confirmed with large prospective cohort studies and, especially, with well-designed controlled, randomized clinical trials (Figure 1).

## 6. Anti-Inflammatory Effects of Nutraceuticals in Experimental Varicocele

For more than 20 years, all studies performed on varicoceles, both in humans and in animals, were meant to contribute to the understanding of the mechanisms involved in the lesion, trying to show the differences between the models and to avoid the risk of considering mechanisms which do not relate to human pathology [52]. Additionally, all papers were meant to demonstrate that surgical repair could counteract all of the known harmful consequences of the varicocele itself [53]. Surprisingly, only at the beginning of this century, were systematic *studies* proposed to evaluate the possible role of different substances, including natural functional foods and nutraceuticals, on the different mechanisms implicated in the pathogenesis of the varicocele. However, an accurate analysis of the literature showed that only a few papers took into account the role of natural functional foods and nutraceuticals to counteract inflammation in varicocele testes (Table 1).

Among these studies, the role of aescin, an extract from *Aesculus hippocastanum* (horse chestnut) seeds, was evaluated. Its biological effects are exerted on vascular endothelial cells, improving venous contraction and increasing blood return, and on the protection of collagen fibers of vein walls and prevention of varicosities. Another important role of aescin is to decrease the activation and migration of leucocytes, thereby reducing inflammation. In experimental varicoceles, lower interstitial edema and reduced polymorphonuclear leucocytes density were observed when aescin was administered [54].

The effects of the polysaccharide of the *Morinda officinalis* How plant (MOP), used as a traditional Chinese medicine, were evaluated in experimental varicoceles in adolescent rats [55]. MOP restored the damaged seminiferous epithelium, upregulated tight junction protein expression in BTB and decreased the levels of TGF-β3 and TNF-α, thus showing a positive effect on the inflammatory pathway.

Other substances whose role in the regulation of structural and functional damages induced by varicoceles were examined are resveratrol and N-Palmitoylethanolamide (PEA). Resveratrol (Res) (3,5,4′-trihydoxy-trans-stilbene) is present in some plants, such as mulberries, peanuts, and grapes, that demonstrate a well-known anti-inflammatory activity. In experimental varicoceles, Res significantly down-regulated the gene expression of NLRP3, a component of the inflammasome complex, modulating the inflammatory pattern [28].

A similar mechanism was demonstrated in PEA, an endogenous compound of living organisms synthesized from precursor phospholipid and found in the seeds of some legumes, in some vegetables, and in milk. In murine varicocele, PEA reduces NLRP3 expression; in addition, its anti-inflammatory mechanism in varicoceles can be explained by the concomitant down-regulation of pErk 1/2 and TGF- β3, thus improving the functional and structural features [56].

The effects of silymarin (SMN) in a rat model of varicoceles were also considered. SMN is an extract from the ripe seeds of *Silybum marianum*; it contains a combination of flavonolignanes, of which silibinin is the principal component. Mazhari et al. [27] showed that the administration of SMN alone was able to decrease the COX2 mRNA level, probably following a reduced production of ROS and NO related to the inflammatory status observed in varicoceles.

The combination of monotropein, astragalin, and spiraeoside obtained from a mixture of herbal plant extracts was also evaluated [2]. Monotropein is a glycoside from MOP, characterized by an anti-inflammatory activity [57], as shown in experimental varicoceles. Astragalin is a flavonoid from *Cuscuta chinensis* Lamark, which is able to improve male reproductive function [58]. Spiraeoside, the predominant flavonoid present in *Allium cepa* L. (Liliaceae), recovered spermatogenesis in experimental varicoceles (Soni et al., 2015 [58]). A treatment with an association of the three substances exhibited an anti-inflammatory effect in experimental varicoceles through the downregulation of proinflammatory cytokines such as TNF-α and IL-6 [2].

Another natural substance with an anti-inflammatory role in experimental varicoceles was shown to be berberine (BBR), an isoquinoline alkaloid found in *Coptis chinensis* and *Hydrastis canadensis* [18]. Treatment with BBR reduced either the infiltration of polymorphonuclear and mononuclear immune cells or the high levels of IL-6 and TNF-α, improving the hormonal balance and the structural organization of the testes [18].

Gui-A-Gra, a commercial powder that has recently been obtained from the edible insect *Gryllus bimarculatus*, has shown pharmacological activity, owing to the presence of an anti-inflammatory glycosaminoglycan [59]. In experimental varicoceles, treatment with Gui-A-Gra significantly reduced the levels of IL-6 and TNF-α, stimulated testosterone biosynthesis, and improved the structural organization of the seminiferous tubules [29].

Furthermore, the effects of an herbal combination in experimental varicoceles following varicocelectomy were examined, and a reduction of IL-6, IL-1β, and TNF-α was demonstrated [60].

**Table 1 ijms-23-16118-t001:** Anti-inflammatory approaches in experimental varicocele.

Animal	Treatment	Results	Reference
Rat	Aescin (20–40 mg/kg intragastric) for 7 weeks after 4 weeks from surgery	Lower interstitial edema Reduced polymorphonuclear leucocytes density	[54]
Rat	*Morinda officinalis* Polysaccharide (200–300–400 mg/kg gavage) for 4 weeks after 8 weeks from surgery	Reduction of TGF-β3 and TNF-α Upregulation of tight junction protein expression	[55]
Rat	Resveratrol (20–50 mg/kg gavage) for one month after 3 months from surgery	Reduction of NLRP3 inflammasome	[28]
Rat	Silymarin (50 mg/kg orally) for 60 days after surgery	Reduction of COX-2	[27]
Rat	MOTILIPERM (monotropein, astragalin, spiraeoside) (200 mg/kg gavage) for 28 days after 4 weeks from surgery	Reduction of IL-6 and TNF-α	[2]
Rat	Berberine (50–100 mg/kg i.p.) for 60 days after surgery	Reduction of IL-6 and TNF-α Diminished immune cells infiltration	[18]
Rat	Gui-A-Gra (1.63–6.5 g/kg orally) for 42 days after 4 weeks from surgery	Reduction of IL-6 and TNF-α	[29]
Rat	Erbal combination (200–400 mg/kg orally) for 4 weeks after surgery	Reduction of IL-6, IL-1β and TNF-α	[60]
Rat	PDRN (8 mg/kg i.p.)-Se (0.4 mg/kg i.p.) for 30 days after 28 days from surgery	Reduction of NLRP3 inflammasome and IL-1β	[13]
Mouse	N-Palmitoylethanolamide (PEA) (10 mg/kg i.p.) for 21 days after 28 days from surgery	Reduction of NLRP3 inflammasome, pERK 1/2, TGF-β3 Improvement of tubular and extratubular organization	[56]

Selenium (Se), an essential element, plays several favorable roles in the human body, among which, an anti-inflammatory effect was proposed [61], as it is the integral compound of selenoproteins, which are able to counteract inflammatory reactions. In fact, in addition to its role in reducing oxidative stress, it was demonstrated that selenoproteins may regulate IL-6 and TNF-α production in macrophages, thus playing a main role in regulating the inflammatory responses. Polydeoxyribonucleotide (PDRN), the active fraction obtained from trout spermatozoa, is an agonist of adenosine A2A receptor, which can interfere with harmful mechanisms observed in low tissue perfusion, such as in varicoceles. The treatment with Se-PDRN association significantly improved testis structural organization, owing to a reduction of NLRP3 inflammasome and IL-1β expression [13], indicating that Se-PDRN may be a new pharmacological tool that can support surgery (Figure 1).

Additionally, many other favorable compounds have been studied in varicoceles. Among them, vitamin C [62,63], vitamin D [64], vitamin E [65,66], coenzyme Q10 [67], L-carnitine [68,69], zinc [70], and magnesium [71] are included. However, each of these substances have shown a primary antioxidant effect in varicoceles, which could also have anti-inflammatory properties, as the link between oxidative stress and inflammation is well-known.

## 7. Conclusions

Recent experimental animal studies and current clinical trials suggest the positive effects of dietary habits on fertility-associated problems in varicoceles. Dietary habits could be related to the presence of functional foods and/or specifical substances with nutraceutical activity that positively impact a series of molecular pathways directly or indirectly linked to exaggerated inflammatory response and deeply demodulated in varicoceles.

Probably, in our opinion, a better experimental knowledge of the mechanisms of actions involved could help to better define a healthy diet and/or specific functional foods and/or clinically useful nutraceuticals, in addition to surgery, for the therapeutic management of fertility-associated problems in varicoceles. Therefore, future research appears fascinating but also complex. In our humble opinion, the development of an appropriate meta-analysis could be useful to improve our knowledge about the intriguing crosstalk between fertility and foods in varicocele patients.

## Figures and Tables

**Figure 1 ijms-23-16118-f001:**
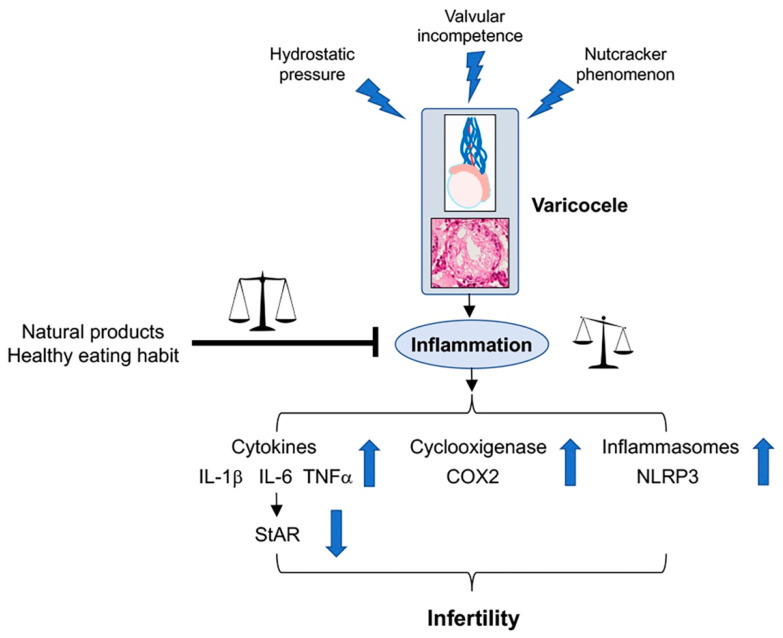
The figure shows the mechanisms able to induce varicocele, the role of the inflammatory cascade on the testis, and the behavior of inflammation mediators responsible for an altered fertility. The inhibitory role of natural products and healthy diet is also shown.

## Data Availability

Not applicable.

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
