# Peer review of "Varicocele, Functional Foods and Nutraceuticals: From Mechanisms of Action in Animal Models to Therapeutic Application"

_ijms, 2022, doi:10.3390/ijms232416118_

Round 1
Reviewer 1 Report
Varicocele remains to be one of the most notable causes for the loss or endangerment of male fertility, it is also one of the most studied etiologies of male reproductive dysfunction, which is why an update on the most recent findings on the management of varicocele are always welcome in the field.
In this paper Marini et al. focus on a nutraceutical approach which has received an increased attention from andrologists over the past years. In this sense, the topic is timely and interesting. Nevertheless, I do have several comments that need to be clarified before I can support publication of this submitted paper:
- It seems to me than the authors strived to collect data exclusively from animal studies, since they mention animal models for this health condition and the provided table summarizes data from rat studies. If so, this needs to be emphasized on (I would also mentioning this in the title and abstract). If not, than the manuscript needs a more substantial and thorough review of literature since there is a multitude of clinical trials on human subjects taking advantage of substances or nutraceuticals.
- Just by doing a quick research, I found that more potentially favorable compounds have been studied in relation to varicocele, such as vitamins (C, D or E), coenzyme Q10, carnitine, zinc or magnesium. These, however are not mentioned in this review. Why?
- If the review is focused on rat models, this section could benefit from a more thorough description of the features as well as pros and cons of each model. Perhaps, a simple illustration of the differences between the rat models could help the reader to understand the specific features of each.
- The manuscript keeps referring to Figure 1, which however is quite simple. How does it address the statement in lines 209-211 (“all observational studies up to now conducted in humans may prove associations but not causation, the associations between dietary patterns, foods and nutrients need to be confirmed with large prospective cohort studies and, especially, with well-designed controlled, randomized clinical trials”) or lines 297-298 (“Se-PDRN may be a new pharmacological tool in support to surgery”)? Also, the down-regulation of StAR during varicocele is linked to the figure, nevertheless this is not even illustrated in the figure. What is actually the role of Figure 1 in this paper?
Minor comments:
- Please, put Latin names and definition in italics
- In the introduction, one sentence represents one paragraph (lines 32-44). This can be easily merged into one comprehensive paragraph. The same applies for lines 157-179.
- Line 118: please number the reference “Pan et al. (2018)”.
Author Response
Reviewer 1
We thank the Reviewer 1 for her/his suggestions aimed at improving the quality of our paper and at adding new, interesting topics.
Varicocele remains to be one of the most notable causes for the loss or endangerment of male fertility, it is also one of the most studied etiologies of male reproductive dysfunction, which is why an update on the most recent findings on the management of varicocele are always welcome in the field.
In this paper Marini et al. focus on a nutraceutical approach which has received an increased attention from andrologists over the past years. In this sense, the topic is timely and interesting. Nevertheless, I do have several comments that need to be clarified before I can support publication of this submitted paper:
- It seems to me than the authors strived to collect data exclusively from animal studies, since they mention animal models for this health condition and the provided table summarizes data from rat studies. If so, this needs to be emphasized on (I would also mentioning this in the title and abstract). If not, then the manuscript needs a more substantial and thorough review of literature since there is a multitude of clinical trials on human subjects taking advantage of substances or nutraceuticals.
As kindly suggested by the Reviewer, we better emphasized in the title and in the abstract of revised version of our manuscript that the new mechanistic data on positive actions of nutraceuticals and/or functional foods, mostly arise from studies in animal models. Of course, we deeply thank the referee for the suggestion since, in light of our experience on experimental animal models together with new information from human subjects, we are planning to better explore, also through the development of an appropriate meta-analysis, this intriguing crosstalk between fertility and foods in varicocele patients. A specific sentence has been added in the conclusions section of revised manuscript
- Just by doing quick research, I found that more potentially favorable compounds have been studied in relation to varicocele, such as vitamins (C, D or E), coenzyme Q10, carnitine, zinc or magnesium. These, however, are not mentioned in this review. Why?
We agree with the Reviewer about the existence of many other compounds which have shown favorable effects on varicocele-induced changes in testes. However, as the aim of our review was to evaluate the molecules whose mechanism of action is mainly anti-inflammatory, the effects of vitamins (C, D or E), coenzyme Q10, carnitine, zinc or magnesium, linked instead to an antioxidant mechanism, was not examined in detail. However, for a better comprehension we added, at the end of the chapter “5. Anti-inflammatory effects of nutraceuticals in varicocele”, a new paragraph about the aforementioned compounds.
- If the review is focused on rat models, this section could benefit from a more thorough description of the features as well as pros and cons of each model. Perhaps, a simple illustration of the differences between the rat models could help the reader to understand the specific features of each.
As kindly requested by the Reviewer, we better explained the futures of varicocele animal models currently utilized.
So far, the surgical approach of Najari and co-workers, appears more effective than classical approach proposed by Saypol et al. Overall, it has been suggested that the latest surgical approach in order of time, allows to better characterize the testicular immune response, including the production of anti-sperm antibodies and inflammatory factors as well as the robust activation of inflammatory pathways in varicocele (see Ref. n°1)
- The manuscript keeps referring to Figure 1, which however is quite simple. How does it address the statement in lines 209-211 (“all observational studies up to now conducted in humans may prove associations but not causation, the associations between dietary patterns, foods and nutrients need to be confirmed with large prospective cohort studies and, especially, with well-designed controlled, randomized clinical trials”) or lines 297-298 (“Se-PDRN may be a new pharmacological tool in support to surgery”)? Also, the down-regulation of StAR during varicocele is linked to the figure, nevertheless this is not even illustrated in the figure. What is actually the role of Figure 1 in this paper?
As suggested by the Reviewer, in the former 211 and 298 lines the reference to Fig. 1 was erased, as it has been indicated for a mistake during typing of the manuscript. In addition, the Figure was improved by adding the downregulation of StAR secondary to the increase in pro-inflammatory cytokines, able to induce, together with the other indicated mechanisms, infertility in varicocele. Furthermore, the role of the Figure 1 was better explained in the legend, placed under the figure.
Minor comments:
- Please, put Latin names and definition in italics
As requested by the Reviewer, all names and definitions were put in Latin, where necessary.
- In the introduction, one sentence represents one paragraph (lines 32-44). This can be easily merged into one comprehensive paragraph. The same applies for lines 157-179.
As suggested by the Reviewer, the paragraphs from previous line 32 to previous line 44 and the paragraphs from previous line 157 to previous line 179 were merged into single paragraphs.
- Line 118: please number the reference “Pan et al. (2018)”.
As suggested by the Reviewer, the reference number was added.

Reviewer 2 Report
This manuscript provides a comprehensive review of varicocele, inflammation, and infertility. Highlighting the involvement of diet, functional foods of natural origin and nutraceuticals, and its anti-inflammatory role in varicocele. The manuscript is clearly written, figures are adequate, and references are relevant. The work is well presented and written in a logical way. I would like to make some suggestions to help improve the manuscript.
Abstract:
- Line 23 change “Aim of” for “The aim of”
Introduction:
- Throughout the introduction there are very short paragraphs, I suggest putting together some paragraphs so that they are not so short.
o The second paragraph (line 35) can be combined with the first (line 34)
o The fourth paragraph (line 41) can be joined with the third (line 40).
Materials and Methods:
I strongly suggest that authors should include a “materials and methods” section to specify all the information that has been omitted. The authors have to include the information related to the methodology they have performed and incorporate the following information about the search strategy and information processing:
· Consulted databases
· Keywords used for the search
· Number of articles obtained
· Inclusion criteria (types of articles, date range...)
· Exclusion criteria
I consider that this information is highly important to know how the selection of the cited articles in this review has been made.
Author Response
Reviewer 2
General comment: This manuscript provides a comprehensive review of varicocele, inflammation, and infertility. Highlighting the involvement of diet, functional foods of natural origin and nutraceuticals, and its anti-inflammatory role in varicocele. The manuscript is clearly written, figures are adequate, and references are relevant. The work is well presented and written in a logical way.
We thank the Reviewer for his/her very favorable opinion on our work.
I would like to make some suggestions to help improve the manuscript.
Abstract:
- Line 23 change “Aim of” for “The aim of”
As suggested by the Reviewer, the change has been made.
Introduction:
- Throughout the introduction there are very short paragraphs, I suggest putting together some paragraphs so that they are not so short.
o The second paragraph (line 35) can be combined with the first (line 34)
o The fourth paragraph (line 41) can be joined with the third (line 40).
As suggested by the Reviewer, both the paragraphs were joined.
Materials and Methods:
I strongly suggest that authors should include a “materials and methods” section to specify all the information that has been omitted. The authors have to include the information related to the methodology they have performed and incorporate the following information about the search strategy and information processing:
- Consulted databases
- Keywords used for the search
- Number of articles obtained
- Inclusion criteria (types of articles, date range...)
- Exclusion criteria
I consider that this information is highly important to know how the selection of the cited articles in this review has been made.
As requested by the Reviewer, a new section “Materials and Methods” was added.

Round 2
Reviewer 1 Report
The authors have addressed my concerns and incorporated by suggestions to the manuscript. I have no further comments.
Reviewer 2 Report
The manuscript has been improved. The authors have incorporated information related to the search strategy in the materials and methods section.